# Intermolecular Halogen Bond Detected in Racemic and Optically Pure N-C Axially Chiral 3-(2-Halophenyl)quinazoline-4-thione Derivatives

**DOI:** 10.3390/molecules27072369

**Published:** 2022-04-06

**Authors:** Ryosuke Matsui, Erina Niijima, Tomomi Imai, Hiroyuki Kobayashi, Akiko Hori, Azusa Sato, Yuko Nakamura, Osamu Kitagawa

**Affiliations:** 1Department of Applied Chemistry (Japanese Association of Bio-Intelligence for Well-Being), Shibaura Institute of Technology, 3-7-5 Toyosu, Kohto-ku, Tokyo 135-8548, Japan; mc21035@shibaura-it.ac.jp (R.M.); mc19015@shibaura-it.ac.jp (E.N.); mc18005@shibaura-it.ac.jp (T.I.); 2Graduate School of Engineering and Science, Shibaura Institute of Technology, 307 Fukasaku, Minuma-ku, Saitama 337-8570, Japan; mc20017@shibaura-it.ac.jp (H.K.); ahori@shibaura-it.ac.jp (A.H.); 3Center for Medical and Nursing Education, Division of Basic Sciences, Tokyo Women’s Medical University, 8-1 Kawada-cho, Shinjuku-ku, Tokyo 162-8666, Japan; sato.azusa@twmu.ac.jp (A.S.); nakamura.yuko@twmu.ac.jp (Y.N.)

**Keywords:** halogen bonds, axial chirality, atropisomers, quinazolinones, thiones

## Abstract

The halogen bond has been widely used as an important supramolecular tool in various research areas. However, there are relatively few studies on halogen bonding related to molecular chirality. 3-(2-Halophenyl)quinazoline-4-thione derivatives have stable atropisomeric structures due to the rotational restriction around an N-C single bond. In X-ray single crystal structures of the racemic and optically pure N-C axially chiral quinazoline-4-thiones, we found that different types of intermolecular halogen bonds (C=S⋯X) are formed. That is, in the racemic crystals, the intermolecular halogen bond between the *ortho*-halogen atom and sulfur atom was found to be oriented in a periplanar conformation toward the thiocarbonyl plane, leading to a syndiotactic zig-zag array. On the other hand, the halogen bond in the enantiomerically pure crystals was oriented orthogonally toward the thiocarbonyl plane, resulting in the formation of a homochiral dimer. These results indicate that the corresponding racemic and optically pure forms in chiral molecules are expected to display different halogen bonding properties, respectively, and should be separately studied as different chemical entities.

## 1. Introduction

Atropisomers arising from the rotational restriction around a biaryl bond are key structural components in chiral catalysts, natural products, biologically active compounds and chiral functional materials [1,2,3,4], and their enantioselective construction has been studied by many groups [5,6]. In contrast to well-known atropisomeric biaryls, atropisomers that emerge due to the rotational restriction around an N-C single bond, namely, N-C axially chiral compounds have attracted scant attention until recently [7]. In 2005 and 2010, we reported highly enantioselective syntheses of *ortho-tert*-butyl anilides (N-C axially chiral amides) and *N*-(*ortho-tert*-butylphenyl)-2-arylindoles (N-C axially chiral nitrogen-containing aromatic heterocycles) through chiral Pd-catalyzed N-C bond forming reactions [8,9]. Since the publication of these reactions, N-C axially chiral compounds have been widely accepted as new target molecules for catalytic asymmetric reactions, and highly enantioselective catalytic syntheses of diverse N-C axially chiral compounds have been achieved (more than one hundred original papers on the catalytic enantioselective synthesis of N-C axially chiral compounds have been published to date) [10,11,12]. Furthermore, the synthetic utility of N-C axially chiral products has been demonstrated through their application in various asymmetric reactions [10,11,12]. At the present time, N-C axially chiral chemistry has become a hot research topic in the field of synthetic organic chemistry.

These N-C axially chiral derivatives are also pharmaceutically attractive compounds. For example, 3-(2-halophenyl)quinazolin-4-one derivatives **1a** and **1b**, which are also known as mebroqualone and mecloqualone, are stable atropisomeric compounds due to the rotational restriction around an N-Ar bond (the rotational barriers = Δ*G*^‡^ = ≥31.5 kcal mol^−1^) and they possess a tranquilizing effect based on GABA receptor agonist activity (Figure 1 and Figure 1) [13,14]. In 2016, we succeeded in the catalytic asymmetric synthesis of axially chiral mebroqualone **1a** and its derivatives [15].

In the course of this study, a unique property of a halogen bond (chirality-dependent halogen bond) was found in the crystals of **1a** and **1b** [16]. As shown in Figure 1, in the crystal of racemic **1a** (and **1b**), an intermolecular halogen bond (C=O⋯X) between the *ortho*-halogen and carbonyl oxygen atoms was formed, leading to the syndiotactic zig-zag array in which molecules of opposite absolute configuration were alternately connected. In contrast, the analogous formation of a halogen bond was not found in the crystals of optically pure **1a** and **1b**. Over the last two decades, halogen bonding (X⋯B) has aroused great interest as a new type of noncovalent interaction and has been widely used as an important supramolecular tool in various fields such as crystal engineering, liquid crystals, material science, and biological systems [17,18,19,20,21,22,23,24]. On the other hand, although halogen bond-mediated chiral recognitions and chiral catalysts with halogen bond donor functionality have also been reported, there are relatively few studies on halogen bonding related to molecular chirality such as those shown in Figure 1 [25,26,27,28,29,30]. The halogen bond observed in axially chiral quinazolinones **1a** and **1b** suggests that in the case of chiral compounds, the corresponding racemic and optically active forms, ought to have different halogen bonding properties, and should be explored as different chemical entities. In addition, since homo- and hetero-chiral associations sometimes affect the pharmacokinetics and drug efficacy [31], the chirality-dependent halogen bond may offer meaningful information to contemplate the difference in pharmacological activity between the optically pure forms and the racemates.

It has been reported that the conversion of quinazolinones to quinazolinone-thione analogues significantly increased their pharmacological activity [32,33,34]. Quite recently, we found that optically pure mebroqualone **1a** and mecloqualone **1b** can be converted to the corresponding thione analogues, i.e. 3-(2-bromophenyl)-2-methylquinazoline-4-thione (**2a**) and 3-(2-chlorophenyl)-2-methylquinazoline-4-thione (**2b**), respectively, without a decrease in enantiomeric excess through the reaction with Lawesson’s reagent (Figure 1) [35]. The obtained thione derivatives **2a** and **2b** have higher rotational barriers than the corresponding quinazolinones **1a** and **1b**. The reaction in Figure 1 is the first example of the conversion of optically active N-C axially chiral amides to thioamide analogues. The sulfur atom of the C=S group is also known to work as a Lewis base in the halogen bond [36,37,38,39,40,41], while it is less commonly studied in comparison with halogen bonding with C=O and C=N functional groups. We were curious about whether the chirality-dependent halogen bond detected in **1a** and **1b** is also observed in crystals of the thione analogues **2a** and **2b**.

In this article, we report the crystal structure and intermolecular halogen bonds (C=S⋯X) which were detected in the crystals of racemic and optically pure N-C axially chiral 3-(2-halophenyl)quinazoline-4-thione derivatives. In contrast to quinazolin-4-one derivatives, in quinazoline-4-thione analogues, the intermolecular halogen bond was detected in not only racemic crystals but also in optically pure crystals. Meanwhile, the halogen bonds in the racemate and optically pure form were found to be oriented in a different direction (angle) toward thiocarbonyl plane.

## 2. Results and Discussion

### 2.1. Crystallization and Structure of ***2***

Single crystals of quinazoline-4-thiones **2a** and **2****b** (racemic and optically pure forms) were prepared through recrystallization by natural evaporation from a solution of hexane and chloroform (1:1) which yielded yellow prismatic crystals suitable for X-ray crystallographic studies. Racemates **2a** and **2b** have an isomorphic relationship with the same Orthorhombic *P*bca space group and have quite similar molecular structures. The structure of **2a** in the racemic crystals is shown in Figure 2b with the corresponding numbering schemes (the structure of quinazolinone **1a** is also shown in Figure 2a for comparison). Selected intramolecular bond distances and angles of racemic **1a**,**b** and **2a**,**b** are summarized in Table 1, showing their similar structure; the bond lengths of C1-S1 for **2a** and **2b** are 1.659(2) and 1.6587(14) Å, respectively. Most of the molecular structural details are similar, but when oxygen is converted to sulfur, the dihedral angle between ring-*A* and ring-*B* of the molecules in the racemic crystals **2a** and **2b** increases to become orthogonal.

### 2.2. Supramolecular Association of ***2a***

The crystal packing and remarkable halogen bond of racemic mebroqualone thione analogue **2a** are shown in Figure 3. The structure of racemic **2a** is very similar to that of racemic mebroqualone **1a** shown in Figure 1. That is, syndiotactic zig-zag arrays, in which molecules of opposite absolute configuration are alternately connected [⋯(*P*)-**2a**⋯(*M*)-**2a**⋯], are formed by the intermolecular interaction between the sulfur and *ortho*-bromine atoms. The intermolecular distance of S⋯Br is 3.4143(6) Å and is 6.5% shorter than the sum of van der Waals radii of bromine and sulfur atoms, which is 3.65 Å. The arrangement of C11-Br1⋯S1^i^ (i: 1/2 − x, 1/2 + y, z) is nearly linear [the bond angle = 172.04(6)°] and this strongly supports the formation of the halogen bond (C=S⋯Br).

In the crystal of optically pure (*P*)-**2a**, although the isotactic arrays were not found, the formation of a homochiral dimer due to the intermolecular interactions between the sulfur and *ortho*-bromine atoms were detected (Figure 4). The intermolecular distance of S⋯Br is 3.422(1) Å and is 6.2% shorter than the sum of van der Waals radii of bromine and sulfur atoms, which is 3.65 Å. This result is remarkably in contrast to optically pure (*P*)-**1a** in which the intermolecular interaction between the carbonyl oxygen and *ortho*-bromo atoms was not detected. The bond angles of C11B-Br1B⋯S1A is 158.2(1)°, showing a somewhat bent arrangement in comparison with that (172.05°) of racemic **2a**; nevertheless, the arrangement would still be within the acceptable range of halogen bonding [40]. Interestingly, the *ortho*-bromine atom in racemic **2a** is located in the periplanar region of the thiocarbonyl group [the torsion angle of C2-C1=S1⋯Br1^ii^ = −19.6(2)°] (ii: 1/2 − x, −1/2 + y, z), while that in (*P*)-**2a** lies in the direction orthogonal to the thiocarbonyl plane [the torsion angle of C-C=S⋯Br is 109.0(3)°]. Thus, the angle of halogen bonds toward the thiocarbonyl plane was revealed to be significantly different in racemic **2a** and (*P*)-**2a**.

The comparative distances and angles of the detected and previously reported halogen bonds are summarized in Table 2. The intermolecular halogen bonds (C=S⋯Br) between achiral substrates, e.g., thiourea and 3-bromopyridine-*N*-oxide (**3**, CCDC1979836) or trithiocyanuric acid and 3-bromopyridine-*N*-oxide (**4**, CCDC2011097) have been previously reported [41,42]. The halogen bond in **3** is oriented orthogonally toward the thiocarbonyl plane [the torsion angle of N-C=S⋯Br is 99.88°] like (*P*)-**2a**, while the torsion angle of N-C=S⋯Br in **4** was 50.95° and −55.90°. The bond angles of S⋯Br-C in **3** and **4** are 168.11° and 172.54–178.97°, respectively and these values are similar to that (172.04°) of racemic **2a**. Although the magnitude of the shortening (2.0%) of halogen bonds in **3** (the distance of S⋯Br is 3.580 Å) was smaller than those (6.5 and 6.2%) in racemic **2a** and (*P*)-**2a,** those (9.1 and 9.3%) in **4** (the distances of S⋯Br are 3.320 and 3.313 Å) were significantly larger in comparison with those in **2a.**

### 2.3. Supramolecular Association of ***2b***

The formation of halogen bonds bearing the different torsion angles (<C-C=S⋯X) detected in racemic **2a** and (*P*)-**2a** was also observed in 2-methylquinazoline-4-thione **2b**, bearing an *ortho*-chlorophenyl group. The X-ray crystal structures of racemic **2b** and optically pure (*P*)-**2b** are shown in Figure 5. Similar to the *ortho*-bromo derivative **2a** and quinazolinones **1a**,**b**, in the crystal of racemic **2b**, syndiotactic zig-zag arrays [⋯(*P*)-**2b**⋯(*M*)-**2b**⋯], which were formed by intermolecular halogen bonds between the sulfur and *ortho*-chloro atoms, were found (Figure 5a). The intermolecular distance of S⋯Cl is 3.4847(6) Å and is 1.8% shorter than the sum of van der Waals radii of chlorine and sulfur atoms, which is 3.55 Å. The bond angle of C11-Cl1⋯S2^i^ is 173.86(5)°, showing the formation of halogen bond (C=S⋯Cl). Similar to racemic **2a**, the chlorine atom is located in the periplanar region of the thiocarbonyl plane [the torsion angle of C-C=S⋯Cl is 20.0(2)°].

In the crystal of optically pure (*P*)-**2b**, similar to (*P*)-**2a**, the formation of a homochiral dimer through the intermolecular interaction between the sulfur and *ortho*-chloro atoms was found (Figure 5b). The intermolecular distance of S⋯Cl is 3.464(1) Å and is 2.4% shorter than the sum of van der Waals radii of chlorine and sulfur atoms, which is 3.55 Å [In optically pure (*P*)-**1b**, no intermolecular O⋯Cl interaction was observed]. The bond angles of C11-Cl1⋯S2^iii^ (iii: x, y, 1 + z) and the torsion angle of C-C=S⋯Cl are 158.2(6)° and 112.3(2)°, respectively. This torsion angle indicates that the halogen bond of (*P*)-**2b** is directed orthogonally toward the thiocarbonyl plane, as observed for (*P*)-**2a**.

Din et al. previously reported the intermolecular halogen bond (C=S⋯Cl) between the *ortho*-chloro and sulfur atoms in 1-(2-chloro-5-nitrophenyl)-4,4,6-trimethyl-3,4-dihydropyrimidine-2(1H)-thione **5** (Table 2, CCDC1419338) [43]. The bond angles of S⋯Cl-C and the torsion angle of N-C=S⋯Cl are 166.7° and −49.28°, respectively and these were values between racemic **2b** (173.86° and 20.0°) and (*P*)-**2b** (156.7 °and 112.3°). The magnitude of the shortening (1.6%) of halogen bond in **5** (the distance of S⋯Cl = 3.493 Å) was slightly smaller than those (1.8 and 2.4%) in racemic **2b** and (*P*)-**2b.** Although Din et al. didn’t mention on the chirality of pyrimidine-2-thione **5** in the literature, the NMR spectra and X-ray crystal structure suggest that **5** may possess stable N-C axially chiral structure like **2b**. Interestingly, in the crystal of **5** (racemate), (*P*)- and (*M*)-isotactic arrays, which consist of (*P*)- and (*M*)-**5**, respectively, were formed by the intermolecular halogen bond (C=S⋯Cl). The formation of isotactic array in racemic **5** is remarkably in contrast to syndiotactic array detected in the crystal of racemic **2b**. On the other hand, such halogen bonding was not found in the crystal of pyrimidine-2-thione **6** bearing *ortho*-bromophenyl group (Table 2, CCDC 1063500).

Since the magnitudes of the shortening of the halogen bonds of *ortho*-chloro derivatives **2b** and (*P*)-**2b** (1.8% and 2.4%, respectively) are smaller than those of *ortho*-bromo derivatives **2a** and (*P*)-**2a** (6.5% and 6.2%, respectively), it is obvious that the S⋯Br interaction is stronger in comparison with the S⋯Cl interaction.

### 2.4. Crystal Packing of **2a** and ***2b***

Figure 6 shows the crystal packings in racemic **2a**,**b** and (*P*)-**2a**,**b**. A crystal lattice of racemic **2a**,**b** is constructed by four (*P*)- and (*M*)-molecules (total eight molecules), and four halogen bonds between heterochiral molecules are formed in each lattice (blue dashed lines). In addition, four heterochiral halogen bonds with molecules in different lattices were found (red dashed lines). In optically pure (*P*)-**2a**,**b**, the crystal lattice consists of four molecules, and each molecule forms homochiral halogen bonds between molecules in the different lattices (red dashed lines). The melting point and crystal density in racemic **2a** and (*P*)-**2a** are not so different, while in the *ortho*-chloro derivative **2b**, those of (*P*)-**2b** are higher than those of racemic **2b**.

### 2.5. Crystal Structures of 3-(2-Fluorophenyl)-2-methylquinazoline-4-thione (***2c***)

We previously reported that 3-(2-fluorophenyl)-2-methylquinazolin-4-one **1c** is a rare atropisomeric compound bearing an *ortho*-fluorophenyl group and the rotational barrier is high enough (Δ*G*^‡^ = 26.1 kcal mol^−1^) for enantiomer separation via a chiral HPLC method (half-life of racemization = *t*_1/2_ = 8.7 days at 298 K) [44]. On the other hand, in the crystal of racemic **1c**, the intermolecular halogen bond (C=O⋯F) between the *ortho*-fluorine and carbonyl oxygen atoms was not found [16]. We converted quinazolinone **1c** to the thione analogue **2c** through the reaction with Lawesson’s reagent (Figure 2) and attempted to conduct X-ray crystal structural analysis. The crystallization of **2c** was achieved through natural evaporation from a solution of hexane and chloroform (1:1) to give yellow prismatic crystals.

Similar to racemic *ortho*-bromo and chloro derivatives **2a**,**b**, the crystal lattice in racemic **2c** is constructed by four (*P*)- and (*M*)-molecules (total eight molecules) as shown in Figure 7. The molecular structures of **2a**–**c** are similar but the dihedral angle between ring-*A* and ring-*B* of the molecules in the crystal **2c** is slightly small, 82.64° in comparison with those in the crystals **2a**,**b** (86.93° and 87.84°, respectively). In the packing structure, the intermolecular π⋯π stackings between quinazolinone rings were found to give the dimer between (*P*)- and (*M*)-**2c**. Typically, the intermolecular distance between two ring-*C* centroids is 3.5335(13) Å [the corresponding perpendicular distance is 3.3672(8) Å], showing the strong stacked structure, and the dimer further interacts between two ring-*A* centroids with a distance of 4.2843(12) Å [the corresponding perpendicular distance is 3.4393(7) Å] to give columnar stacking along the *a* axis. The weak C-H⋯π and CH⋯F interactions were also observed in the packing. Unfortunately, the intermolecular halogen bond (C=S⋯F) between *ortho*-fluoro and sulfur atoms was not detected. It is well known that the order of strength of the halogen bond is I⋯B > Br⋯B > Cl⋯B >> F⋯B and that halogen bonds (F⋯B) between fluorine atoms and a Lewis base are uncommon.

To understand all the intermolecular interactions, the Hirshfeld surface (HS) analysis [45] was carried out for the crystals of **2a**–**c** using Crystal Explorer 17.5 and the results are summarized in Appendix A. The main interactions for the whole structure of racemic **2a** are H(inside)⋯H(outside) and C⋯H/H⋯C contributing 36.5% and 24.7%, respectively, to the overall crystal packing due to the high surface area of the aromatic. The presence of two halogen bonds are found to be the contributions of the S⋯Br (1.2%) and Br⋯S (1.2%) contacts with the sharp finger plots, indicating the 1D alternate zig-zag arrangement. No contribution of the C⋯C (1.0%), Br⋯Br (0%), and S⋯S (0%) shows the predominant halogen bonds. The N⋯H/H⋯N contribution is also observed 5.3% between the quinazoline rings. On the other hand, the results of (*P*)-**2a** shows only one-way halogen bonds and the corresponding contributions of the S⋯Br/Br⋯S is 1.0% with the sharp finger plot. The contributions of H⋯H, C⋯H/H⋯C, C⋯C, Br⋯Br, S⋯S, and N⋯H/H⋯N are 35.6%, 27.8%, 0.8%, 1.1%, 0%, and 4.7%, respectively. The characteristics of the interaction of racemic **2a** and (*P*)-**2a** are also reflected well in chloro-derivatives **2b**, but the halogen bonds of **2a** are more effective; the S⋯Cl/Cl⋯S for racemic **2b** and (*P*)-**2b** are 2.1% and 0.9%, respectively. The HS analysis of **2c** shows the weak intermolecular interactions and the corresponding contribution of π⋯π, C-H⋯π, and H⋯F interactions through C⋯C (5.4%), C⋯H/H⋯C (23.4%), and H⋯F/F⋯H (9.4%), respectively. The contributions of S⋯F/F⋯S and N⋯H/H⋯N are 1.8% and 7.1%, respectively, on the HS but no direct halogen bonds were detected from the finger plots.

We also succeeded in the synthesis of optically pure 2-methylquinazoline-4-thione bearing an *ortho*-iodophenyl group, through the reaction shown in Figure 1. However, since the *ortho*-iodo derivative is extremely unstable, gradual decomposition occurs during the crystallization and the preparation of a single crystal was not possible.

## 3. Materials and Methods

### 3.1. General Information

The melting points were uncorrected. The ^1^H and ^13^C NMR spectra were recorded on a 400 MHz spectrometer. In the ^1^H and ^13^C NMR spectra, chemical shifts were expressed in δ (ppm) downfield from CHCl_3_ (7.26 ppm) and CDCl_3_ (77.0 ppm), respectively. HRMS were recorded on a double focusing magnetic sector mass spectrometer using electron impact ionization. Column chromatography was performed on silica gel (75–150 μm). Medium-pressure liquid chromatography (MPLC) was performed on a 25 × 4 cm i.d. prepacked column (silica gel, 10 μm) with a UV detector. High-performance liquid chromatography (HPLC) was performed on a 25 × 0.4 cm i.d. chiral column with a UV detector.

### 3.2. Racemic and Optically Pure 3-(2-Halophenyl)-2-methylquinazoline-4-thiones ***2a***–**c**

Racemic and optically pure forms of 3-(2-bromophenyl)-2-methylquinazoline-4(3*H*)-thione (**2a**) and 3-(2-chlorophenyl)-2-methylquinazoline-4(3*H*)-thione (**2b**) were prepared in accordance with the synthetic procedure that we previously reported [15,16,35].

3-(2-Fluorophenyl)-2-methylquinazoline-4(3H)-thione (**2c**). Under N_2_ atmosphere, 3-(2-fluorophenyl)-2-methylquinazolin-4(3*H*)-one **1c** [44] (76 mg, 0.3 mmol) and Lawesson’s reagent (182 mg, 0.45 mmol) in xylene (5 mL) were stirred for 3 h under xylene reflux conditions (oil bath at 150 °C). The mixture was poured into water and extracted with AcOEt. The AcOEt extracts were washed with brine, dried over MgSO_4_, and evaporated to dryness. Purification of the residue by column chromatography (hexane/AcOEt = 6) gave racemic **2c** (77 mg, 95%). **2c**: yellow solid; mp 110–112 °C (racemate); IR (neat) 1595, 1497 cm^−1^; ^1^H NMR (400 MHz, CDCl_3_) δ: 8.74 (1H, dd, *J* = 1.2, 7.9 Hz), 7.80 (1H, td, *J* = 1.2, 6.7 Hz), 7.69 (1H, dd, *J* = 1.2, 8.6 Hz), 7.48–7.57 (2H, m), 7.28–7.39 (3H, m), 2.32 (3H, s); ^13^C NMR (100 MHz, CDCl_3_) δ: 190.3, 157.1 (d, *J*_C–F_ = 249.8 Hz), 153.1, 142.7, 135.4, 131.8 (d, *J*_C–F_ = 7.6 Hz), 131.5, 130.0 (d, *J*_C–F_ = 13.3 Hz), 129.7, 128.7, 128.3, 127.8, 125.9 (d, *J*_C–F_ = 3.8 Hz), 117.5 (d, *J*_C–F_ = 19.1 Hz), 25.0; MS (*m*/*z*) 271 (MH^+^); HRMS. Calcd for C_15_H_12_FN_2_S (MH^+^) 271.0705. Found: 271.0713.

### 3.3. X-ray Single Crystal Structural Analysis

The single crystal X-ray structures were determined by a Bruker D8 Quest with Mo*K*α radiation (*λ* = 0.71073 Å) generated at 50 kV and 1 mA. All the crystals were coated with paratone-N oil and measured at 100 K. SHELXT program was used for solving the structures [46]. Refinement and further calculations were carried out using SHELXL [47]. The crystal data and structure refinement of four crystals are summarized in Table 3. H atoms were placed in geometrically idealized positions and refined as riding, with aromatic C-H = 0.95 Å and *U*_iso_(H) = 1.2*U*_eq_(C). The chiral crystal (*P*)-**2b** shows the correct absolute structure and the flack parameter is −0.02(3). CCDC 2153821 (*rac*-**2a**), 2153822 (*rac*-**2b**), 2153823 [(*P*)-**2b**], and 2153825 (*rac*-**2c**) contain the Appendix A for this paper. These data can be obtained free of charge via http://www.ccdc.cam.ac.uk/conts/retrieving.html, accessed on 3 February 2022. The structural information of CCDC1561609 (*rac*-**1a**), 1897963 (*rac*-**1b**), and 2032428 [(*P*)-**2a**] was used [16,35].

## 4. Conclusions

We found that in the crystals of racemic and optically pure N-C axially chiral 3-(2-bromo- and 2-chloro-phenyl)-2-methylquinazoline-4-thiones, different types of intermolecular halogen bonds between *ortho*-halogens and sulfur atoms are formed. The halogen bonds detected in the racemic crystals are located in the periplanar region of the thiocarbonyl plane, leading to the syndiotactic zig-zag array in which molecules of opposite absolute configuration are alternately connected. The modes of halogen bonding and association are very similar to those detected in the corresponding racemic carbonyl analogues (mebroqualone and mecroqualone). On the other hand, the halogen bonds in the optically pure crystals are oriented orthogonally toward the thiocarbonyl plane, resulting in the formation of a homochiral dimer. This is in remarkable contrast to the optically pure carbonyl analogues in which no intermolecular interaction between carbonyl oxygen and *ortho*-halogen atoms was detected. Although it is well known that the different types of hydrogen bonding are formed in racemic and enantiomerically pure crystals [48,49], homo- and hetero-chirality dependent halogen bonds mentioned in this paper are quite rare [16]. These results clearly expose the fact that the corresponding racemic and enantiomerically pure forms of chiral substances are expected to display different halogen bonding properties and should be separately studied as different chemical entities.

## Data Availability

Not applicable.

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
