# Peer review of "Intermolecular Halogen Bond Detected in Racemic and Optically Pure N-C Axially Chiral 3-(2-Halophenyl)quinazoline-4-thione Derivatives"

_molecules, 2022, doi:10.3390/molecules27072369_

Round 1

Reviewer 1 Report

This manuscript examines intermolecular halogen bond formation with racemic and optically pure N-C axially chiral quinazoline-4-thiones.  Interestingly, these compounds form different types of halogen bonding geometries whether they were enantiomerically pure versus racemic.  In the optically pure form, the halogen bond was oriented orthogonal to the thiocarbonyl, affording a homochiral dimer.  While the racemic crystals preferred to form the intermolecular halogen bond in a periplanar conformation between the ortho halogen atom and the sulfur.  These unexpected results indicate that racemic and optically pure forms of axial chiral molecules behave very differently. 

The halogens in these systems should be very weak halogen bond donors as there are no electron withdrawing groups on the ring. It is quite surprising that there is such a large differentiation between the racemic and enantiopure compounds.   It would be helpful to the readers to add a comparison of this hydrogen bonding to literature examples.  One place where this addition might be appropriate is on page 5 line 152, page 6 the last paragraph (lines 204-207) or in the concluding section.  The authors use “6.5% shorter than the sum of the van der Waals radii of bromine and sulfur atoms” (3.65 versus 3.41 Å) as a way to assess the strength of the halogen bonding interaction.   Please add a typical Br ….S halogen bond length from the literature with a reference here. 

Do computations suggest this is really a halogen bond?  Ie lone pair of Sulfur interacting with a sigma hole on the halide?  Does it have significant charge transfer? 

Author Response

(Reply to Reviewer-1)

Thank you very much for your review and meaningful comments. Our replies to your comments are as follows.

  1. It would be helpful to the readers to add a comparison of this hydrogen bonding to literature examples.  One place where this addition might be appropriate is on page 5 line 152, page 6 the last paragraph (lines 204-207) or in the concluding section.

(Reply)

As this reviewer pointed out, it has been often found that different types of hydrogen bonding are formed in racemic and optically pure crystals. Therefore, the sentence about this point and the relevant literatures were added to the Concluding Section and References (48 and 49), respectively.

  1. The authors use “6.5% shorter than the sum of the van der Waals radii of bromine and sulfur atoms” (3.65 versus 3.41 Å) as a way to assess the strength of the halogen bonding interaction.   Please add a typical Br ….S halogen bond length from the literature with a reference here. 

(Reply)

In accordance with the reviewer’s comment, the data (bond lengths, bond angles, torsion angles) on the halogen bonds (C=S···Br and C=S···Cl), which were previously reported in other literatures, were added to the text (Table 2). We added new table (Table 2 in page 5) to the text. Furthermore, the strength of the halogen bonding interaction between known compounds and our compounds was compared. In addition, the relevant literatures were added to References [41-43].

  1. Do computations suggest this is really a halogen bond?  Ie lone pair of Sulfur interacting with a sigma hole on the halide?  Does it have significant charge transfer?

(Reply)

On the basis of bond angles (the nearly linear arrangement) of S···X-C and the comparison with data in the relevant literatures (Table 2 in page 5), I believe that C=S···X interactions are due to halogen bonding. 

Reviewer 2 Report

This is an example of an excellent work devoted to the current topic of the study of halogen bonds and their possible application for the synthesis of optically pure compounds. In this work, a detailed analysis of supramolecular interactions in crystalline structures has been carried out.

The authors are requested to add to the text of the article a comparative table with the data of the detected halogen bonds for the convenience of readers.

Author Response

(Reply to Reviewer-2)

Thank you very much for your review and meaningful comments. Our reply to your comment is as follows.

  1. The authors are requested to add to the text of the article a comparative table with the data of the detected halogen bonds for the convenience of readers.

(Reply)

In accordance with the reviewer’s suggestion, a comparative table with the data of the detected halogen bonds were added to the text (Table 2 in page 5).    

Reviewer 3 Report

This work by Matsui and co-workers presents the halogen bonding behaviors of a series of 3-(2-halophenyl)quinazolin-4-one compounds (with C=O, 1a=Br, 1b=Cl, 1c=F) and 3-(2-halophenyl)quinazolin-4-thione (with C=S, 2a=Br, 2b=Cl, 2c=F). Via characterizing the X-ray structures of related crystals of either the racemic type (having P-/M- chiralities) and optically pure type, the authors observed different halogen bonding patterns. 

They found compounds 1a,1b,2a and 2b can form very unique alternating zig-zag pattern in the racemic co-crystals but not in the optically pure counterparts. 

This finding is in general interesting but a few points need to be taken care of.

1) The P-/M- chirality in this case can be seen as a hindered torsion rotation. The 180-degree rotation of the halogenated benzene ring is the main reason why this interesting zig-zag pattern is observed. 

Therefore, the suggestion is that the author may do some further literature search to see if similar pattern was reported caused by the torsion difference. 

2) In lines 64-65 of Introduction, it would be good if the author can summarize the previous studies. This may be incorporated to point (1). 

3) What if the methyl group was replaced by hydrogen atom, will the barrier be reduced to allow the free rotation of the C-N bond? 

4) It would be nice if the authors can put more reported data from other literature about the same type of halogen bonds found in this work to do some comparison. 

Author Response

(Reply to Reviewer-3)

Thank you very much for your review and meaningful comments. Our replies to your comments are as follows. 

  1. The P-/M- chirality in this case can be seen as a hindered torsion rotation. The 180-degree rotation of the halogenated benzene ring is the main reason why this interesting zig-zag pattern is observed. Therefore, the suggestion is that the author may do some further literature search to see if similar pattern was reported caused by the torsion difference.
  2. In lines 64-65 of Introduction, it would be good if the author can summarize the previous studies. This may be incorporated to point (1).
  3. It would be nice if the authors can put more reported data from other literature about the same type of halogen bonds found in this work to do some comparison.

(Reply)

We appreciate your meaningful comments 1), 2) and 3). Although we conducted the literature survey in detail, the homo- and hetero-chirality dependent halogen bonding mentioned in this paper could not be found in not only atropisomeric compounds but also centrally chiral compounds. Hence, we didn’t mention about other literatures. This indicates the novelty of the present study. On the other hand, we found the literature on halogen bonding (C=S···Cl) of N-(2-chlorophenyl)pyrimidine-2-thione. Although the NMR and X-ray crystal data suggested that this compound has stable atropisomerism due to the rotational restriction around an N-Ar axis, the authors didn’t mention on the axial chirality at all. We confirmed the halogen bonding in pyrimidine-2-thion and discussed on the intermolecular interaction. In addition, the data and literature on the halogen bonding in pyrimidine-2-thione were added to Table 2 and References 43, respectively (page 6-7).

  1. What if the methyl group was replaced by hydrogen atom, will the barrier be reduced to allow the free rotation of the C-N bond?

(Reply)

As this reviewer mentioned, since 3-aryl-2-non-substituted-quinazolin-4-one derivatives are rotationally unstable, we used 2-methylquinazolin-4-ones in this study.

Reviewer 4 Report

Dr Kitagawa and co-authors presented a series of 3-(2-halophenyl)-3-methylquinazolin-4-thiones bearing an N-C chiral axis. The intermolecular halogen bonded interactions in the single crystal structures of both racemic and optically pure compounds were described and compared in detail. The results in this work are interesting and the publication in Molecule is recommended by this reviewer after the authors considering the following aspects.

  1. In a general manner, in the course of describing the structure of molecular crystals, one first introduces the molecular structures and the intermolecular interactions before telling the crystal packing. Then, it is suggested to move the section 2.2 to after section 2.4.
  2. In section 2.5 (page 7, line 213), the sentence "On the other hand, in the crystal of racemic 1c, the intermolecular halogen bond (C=O···F), which was detected in racemic 1a and 1b, was not found" is somewhat confusing, because there is no C=O...F halogen bond either in 1a or 1b (they show C=O...Br and C=O...Cl halogen bonds, respectively).
  3. In addition to the halogen bonds, is there any other intermolecular interactions in these structures? It will be helpful to visually analyze such intermolecular forces and contacts by Hirshfeld surface, which can be easily done with the software of CrystalExplorer.
  4. The melting points were given for these racemic and optically pure compounds, how are they related to the structures? A more detailed discussion for this is necessary.
  5. The authors should provide the powder X-ray diffraction patterns of these compounds in the supplementary information.

Author Response

(Reply to Reviewer-4)

Thank you very much for your review and meaningful comments. Our replies to your comments are as follows.

  1. In a general manner, in the course of describing the structure of molecular crystals, one first introduces the molecular structures and the intermolecular interactions before telling the crystal packing. Then, it is suggested to move the section 2.2 to after section 2.4.

(Reply)

In accordance with this reviewer comments, the section on crystal packing (2.2) was moved after (2.4) the sections on intermolecular interactions. In addition, the numbers of Figure 3-6 were changed.    

  1. In section 2.5 (page 7, line 213), the sentence "On the other hand, in the crystal of racemic 1c, the intermolecular halogen bond (C=O···F), which was detected in racemic 1a and 1b, was not found" is somewhat confusing, because there is no C=O...F halogen bond either in 1a or 1b (they show C=O...Br and C=O...Cl halogen bonds, respectively).

(Reply)

As you pointed out, since this sentence may cause a misunderstanding, “which was detected in racemic 1a and 1b” was deleted.

  1. In addition to the halogen bonds, is there any other intermolecular interactions in these structures? It will be helpful to visually analyze such intermolecular forces and contacts by Hirshfeld surface, which can be easily done with the software of CrystalExplorer..

(Reply)

No other remarkable intermolecular interactions were observed for crystals 2a-b using Platon and HS analyses. We revised the main text and Supplementary Materials to give the HS information with the corresponding finger plots.

  1. The melting points were given for these racemic and optically pure compounds, how are they related to the structures? A more detailed discussion for this is necessary.

(Reply)

We also thought the relation between the melting points and the structures. In ortho-bromo derivative 2a, the melting point of racemate was slightly higher than that of optically pure form. In contrast, in ortho-chloro derivative 2b, the melting of racemate was considerably lower in comparison with that of optically pure form. Thus, since the correlation between the melting points and the structures could not be found, we didn’t mention in the text. We would appreciate your understanding.    

  1. The authors should provide the powder X-ray diffraction patterns of these compounds in the supplementary information.

(Reply)

From the appearance and SC-XRD studies, it is confirmed that the crystal does not take polymorphism.